# The Design and Optimization of Ceramide NP-Loaded Liposomes to Restore the Skin Barrier

**DOI:** 10.3390/pharmaceutics15122685

**Published:** 2023-11-27

**Authors:** Hümeyra Şahin Bektay, Ali Asram Sağıroğlu, Kübra Bozali, Eray Metin Güler, Sevgi Güngör

**Affiliations:** 1Department of Pharmaceutical Technology, Faculty of Pharmacy, Istanbul University, Istanbul 34116, Türkiye; 2Health Science Institute, Istanbul University, Istanbul 34126, Türkiye; 3Department of Pharmaceutical Technology, Faculty of Pharmacy, Bezmialem Vakıf University, Istanbul 34093, Türkiye; 4Department of Pharmaceutical Technology, Faculty of Pharmacy, Istanbul University-Cerrahpaşa, Istanbul 34500, Türkiye; 5Department of Medical Biochemistry, Faculty of Hamidiye Medicine, University of Health Science, Istanbul 34668, Türkiye

**Keywords:** ceramide, liposome, skin barrier function, experimental design, topical administration, response surface methodology, central composite design

## Abstract

The impairment of skin integrity derived from derangement of the orthorhombic lateral organization is mainly caused by dysregulation of ceramide amounts in the skin barrier. Ceramides, fatty acids, and cholesterol-containing nano-based formulations have been used to impair the skin barrier. However, there is still a challenge to formulate novel formulations consisting of ceramides due to their chemical structure, poor aqueous solubility, and high molecular weight. In this study, the design and optimization of Ceramide 3 (CER-NP)-loaded liposomes are implemented based on response surface methodology (RSM). The optimum CER-NP-loaded liposome was selected based on its particle size (PS) and polydispersity index (PDI). The optimum CER-NP-loaded liposome was imagined by observing the encapsulation by using a confocal laser scanning microscope (CLSM) within fluorescently labeled CER-NP. The characteristic liquid crystalline phase and lipid chain conformation of CER-NP-loaded liposomes were determined using attenuated total reflectance infrared spectroscopy (ATR-IR). The CER-NP-loaded liposomes were imagined using a field emission scanning electron microscope (FE-SEM). Finally, the in vitro release of CER-NP from liposomes was examined using modified Franz Cells. The experimental and predicted results were well correlated. The CLSM images of optimized liposomes were conformable with the other studies, and the encapsulation efficiency of CER-NP was 93.84 ± 0.87%. ATR-IR analysis supported the characteristics of the CER-NP-loaded liposome. In addition, the lipid chain conformation shows similarity with skin barrier lipid organization. The release pattern of CER-NP liposomes was fitted with the Korsmeyer–Peppas model. The cytotoxicity studies carried out on HaCaT keratinocytes supported the idea that the liposomes for topical administration of CER-NP could be considered relatively safe. In conclusion, the optimized CER-NP-loaded liposomes could have the potential to restore the skin barrier function.

## 1. Introduction

The skin barrier function is mainly characterized by stratum corneum lipids consisting of ceramides, fatty acids, and cholesterol. In favor of a significant equimolar ratio of ceramides, fatty acids, and cholesterol, the *stratum corneum* expresses an orthorhombic lateral organization, which provides skin integrity [1]. In atopic dermatitis (AD), there is an impairment of skin integrity derived from derangement of orthorhombic lateral organization caused by dysregulation of ceramides, fatty acids, and cholesterol ratio [2,3,4,5]. The disease is manifested by the downregulation of ceramide biosynthesis [6,7,8]. The Cer-NP, which is N-acylated phytosphingosine having the D-erythro structure linked to normal saturated or unsaturated fatty acid is the main ceramide subtype in a healthy stratum corneum layer and decreases in AD [9]. The stratum corneum lipidomic analysis revealed that CER-NP is reduced in AD patients [10]. Hereby, the extraordinary arrangement of stratum corneum lipids disorganization ends up with impaired skin barrier as well as immunological responses [11].

Even with the usage of topical corticosteroids and calcineurin inhibitors as the first option for the treatment of the disease, the formulations consisting of ceramides have recently attracted attention as lipid replacement therapy for AD [12,13]. With this approach, regaining decreased ceramides to stratum corneum is aimed at restoring the skin barrier function [14,15]. In lipid replacement therapy, the impaired skin barrier is replenished via conventional formulations [16,17,18]. In the market, moisturizers are comprised of ceramides present in varied dosage forms such as serum, face mask, lotion, spray, etc. The main idea of using ceramide comes from repairing the skin barrier by replacing the skin barrier components [19,20]. Dermocosmeceutical products mainly contain ceramides with fatty acids and cholesterol in approximate ratios (Triple Lipid Restore, SkinCeuticals; TriXera, Avène; Ceramidin^TM^, Dr. Jart+). On the other hand, conventional formulations may have a limitation with penetration through the skin’s outermost layer [21]. Therefore, novel carrier systems such as nanoemulsion, solid lipid nanoparticles, and special vesicles have become a trend in the market instead of conventional formulations (CeraVe^®^, Loreal, New York, NY, USA). Furthermore, a controlled release designed system comes to the forefront due to maintaining the replacement of ceramide. The liposomes look promising for the market and are considered the most appropriate and biocompatible nano-formulation through the phospholipid basement, which is the main component of the skin cell membrane. The schematic presentation of ceramide replacement therapy with liposomes is provided in Figure 1.

During the design of dosage forms, numerous aspects of pharmaceutical technology that affect the formulation’s quality should be taken into consideration. In a conventional approach, optimization is carried out by manipulating one factor while holding the rest of the parameters invariable. However, this approach necessitates numerous formulation practices. Formulation development is a time-consuming and over-costing process. Therefore, RSM has gained interest over the past decade. In this approach, the relationship between the variables can be evaluated in detail in a short time while the impact of factors on the formulation is examined by changing multiple parameters at once.

In this study, the design and optimization of CER-NP-loaded liposomes were carried out by RSM. For the optimization of CER-NP-loaded liposomes, it is crucial to limit the amount of cholesterol to maintain the surface characteristics of liposomes. The high hydrophobicity of CER-NP poses a challenge to encapsulate and disperse in aqueous media, which is why the optimization of the amount of oleic acid is essential. By this design, the CER-NP-loaded liposomes were formed in an optimum amount of the CER-NP, oleic acid, and cholesterol. Within this approach, the formulation optimization process was carried out based on the quality parameters of CER-NP-loaded liposomes. The predicted formulation is characterized via dynamic light scattering (DLS) and confocal laser scanning microscope techniques and evaluated by in vitro release test and MTT assay.

## 2. Materials and Methods

### 2.1. Materials

CER-NP was kindly provided by Evonik Industries AG (Essen, Germany). Oleic acid and PEG 400 were gifted from Croda International Prc. (Snaith, UK). Cholesterol was purchased by Sigma-Aldrich Chemie (Ann Arbor, MI, USA). Mannitol SD 200 was gifted from Roquette Freres SA (Lestrem, France). Lipoid S100 was kindly provided by Lipoid GmBH (Ludwigshafen, Germany). NBD-Ceramide (NBD-Cer) was purchased from ABP Bioscience (Rockville, MD, USA). HPLC-grade methanol (MeOH), tetrahydrofuran (THF), and acetonitrile (ACN) were purchased from Merck KGaA (Darmstadt, Germany). Cellulose acetate membrane was purchased from Sartorius (Sartorius AG, Göttingen, Germany). The total assays were performed using ultrapure Milli-Q water (Bedford, MA, USA).

### 2.2. Preparation Method of CER-NP-Loaded Liposomes

The CER-NP-loaded liposomes were prepared with a thin-film hydration method [22]. Briefly, phospholipid was weighed at specific amounts, and CER-NP, cholesterol, and oleic acid were weighed at various concentrations, as shown in Table 1. The lipids are dissolved in a mixture of chloroform/methanol (1:1, *v*/*v*) [23]. Then, the solution was evaporated under a 100-mbar vacuum at 45 °C temperature. After the evaporation, the thin-film layer in the round-bottom flask was hydrated with phosphate-buffered saline (PBS, pH 7.36). The formulation was sonicated at 20 kHz of frequency. Afterward, the prepared CER-NP-loaded liposome was characterized in terms of particle size, PDI value, and zeta potential (ZP) by using ZetaSizer Nano ZS (Malvern, UK). The empty liposomes were prepared with cholesterol, oleic acid, and phospholipid by the abovementioned method but without Cer-NP in their content.

### 2.3. Experimental Design of CER-NP-Loaded Liposomes

In the basics of the experimental design, the independent variables are specified based on the literature [24,25]. The independent variables are specified as the amounts of cholesterol, oleic acid, and CER-NP with 5 variables (−α, −1, 0, +1, +α) based on *central composite design* [26]. The experimental design was obtained by the Design of Expert 12. The selected variables are provided in Table 1. The formulation was evaluated based on two crucial responses: particle size and PDI value.

### 2.4. Characterization of CER-NP-Loaded Liposomes

#### 2.4.1. Measurement of Particle Size and Polydispersity Index and Zeta Potential

The particle size, PDI value, and zeta potential of CER-NP liposomes were measured based on dynamic light scattering principles through ZetaSizer Nano ZS (Malvern Panalytical, Malvern, UK). The samples were diluted with PBS (dilution factor: 3.10^−2^). The measurement was performed with three replicates at 25 °C room temperature.

#### 2.4.2. Determination of HPLC Method of CER-NP in Liposomes

The samples on encapsulation efficiency, drug content, and the release study were analyzed by HPLC. The Cer-NP was analyzed by reversed-phase HPLC system (Prominence LC-20A, Shimadzu Corporation, Kyoto, Japan) equipped with UV–Vis detector (SPD-20A, Shimadzu Corporation, Kyoto, Japan). The method was modified based on the literature. Briefly, the C_18_ RP 150 mm × 4.6 mm, 5 μm column is selected as the stationary phase [26,27]. The mobile phase was prepared with methanol: acetonitrile (3:2) after filtering and degassing. The analysis was performed at isocratically at 0.8 mL/min flow rate with 10 μL injection volume under 45 °C oven temperature. The UV–Vis detector was set to 210 nm to detect the Cer-NP effectively. The validation steps were performed based on ICH Q2 (R2) [28,29]. The standard solutions of linearity were prepared by dilution of stock solution with MeOH:THF (1:1). The other lipid ingredients did not show any interference with Cer-NP peaks. The selectivity, linearity, accuracy, precision (repeatability inter-day and intra-day), limit of detection (LOD), and limit of quantification (LOQ) were calculated. The retention time was 9.27 min. The determination coefficient of the linearity curve (r^2^) was higher than 0.99. The LOD and LOQ values were 1.48 µg/mL and 4.5 µg/mL, respectively.

#### 2.4.3. Encapsulation Efficiency and CER-NP Content

The encapsulation efficiency and CER-NP content of the optimum liposome were determined by centrifugation and HPLC analysis. For CER-NP content samples, the specific amount of liposome was dissolved in the MeOH: THF solvent system, and the liposomal membrane was disrupted. For encapsulation efficiency samples, the liposomes were centrifuged for 30 min at 15,000 rpm (Allegra X-30, Beckman Coulter, Brea, CA, USA). Then, the supernatant, including non-encapsulated CER-NP and other ingredients, was withdrawn and dissolved in the MeOH: THF solvent system. The samples were filtered by membrane filter (PTFE, 0.45 µm). Then, the samples were analyzed by the validated HPLC method. The CER-NP content was calculated based on the theoretical and practical weight of CER-NP. The encapsulation efficiency was calculated based on Equation (1).
EE% = (M_i_/M_t_) × 100(1)

Equation (1): Encapsulation efficiency percentage equation based on sediment, supernatant, and total amounts of CER-NP. EE: encapsulation efficiency; M_i_: the amount of CER-NP encapsulated in a liposome; M_t_: the amount of theoretical CER-NP in the formulation.

#### 2.4.4. CLSM Imagining of CER-NP-Loaded Liposomes

NBD-CER ((N-{6-[(7-nitro-2,1,3-benzoxadiazol-4-yl)amino]hexanoyl}sphingosine) which is a fluorescent agent via nitro group was used for examination to localize the ceramide molecules on the liposomal membrane. The various molar concentrations of NBD-Cer were added to the system to observe which ratio of NBD-Cer was effectively encapsulated in liposomes. Fluorescent liposomes were prepared with ratios of 1.73, 1.39, 0.68, 0.34, and 0.17‰ mmol NBD-Cer [30]. The imagining was captured on a Leica TCS SPE confocal microscope (Leica Microsystem, Wetzlar, Germany) with ACS APO 63×/1.30 oil CS. The spectrum was scanned between 488 and 560 nm. The observation was performed at 25 °C temperature and 40% relative humidity. The images were obtained by five replicates.

#### 2.4.5. ATR-IR Spectroscopy Analysis of CER-NP-Loaded Liposomes

The ATR-IR spectroscopy was carried out with NBD-CER containing optimized liposomes in comparison to the empty liposome at 20 runs. Lipoid S100, cholesterol, and oleic acid were analyzed to control the functional group region specifically. The wavelength was in the range of 400–4000 cm^−1^. The transmittance was observed at a percentage in the range of 0–100. The IRAffinity-1S Fourier Transform Infrared Spectrophotometer (Shimadzu Corporation, Kyoto, Japan) was used in the analysis. The spectral data were evaluated in LabSolutions IR Software (Shimadzu Corporation, Kyoto, Japan).

#### 2.4.6. FE-SEM Imagining of CER-NP-Loaded Liposomes

The optimum CER-NP-loaded liposome was dispersed in a 4% mannitol solution as a cryoprotectant. The dispersion system was lyophilized in *Scanvac Lyophilizer* (Labogene, Lillerød, Denmark) for 24 h. The powder was applied on aluminum cylinder sample stubs (12.7 diameter, 1 mm height) covered with double-sided carbon tape. The dust on the samples was removed by blowing air with an air pump. The optimum CER-NP-loaded liposome was imagined via Apreo2S FE-SEM (Thermo Fischer Scientific, Waltham, MA, USA) with a T1 segmented lower in-lens detector. The imagining was carried out under a 1 kV high vacuum without an Au/Pd conductive coat (FE-SEM Mag: 10,000-100,000×). The FE-SEM imagining of the optimum CER-NP-loaded liposome.

#### 2.4.7. Solubility Studies of CER-NP

The medium of the in vitro release test was specified based on the solubility test of CER-NP in various release media (R1: PBS; R2: 10% propylene glycol; R3: 10% PEG 400; R4: 2% Tween 80; R5: 1% bovine serum albumin (BSA); R6: 25% ethyl alcohol) to provide sink condition. Briefly, an excess amount of CER-NP was added to the release medium and kept in a shaker for 72 h under room temperature. The samples were centrifuged, and the supernatant was dissolved in a solvent and quantified by the validated HPLC method.

#### 2.4.8. In Vitro Release of CER-NP from Liposomes

The amount of receptor phase sample was specified according to the amount of dissolved CER-NP in selected receptor media. The effective diffusion area was 1.77 cm^2^, and 0.5 mL volume was used. The receptor media (R3: 10% PEG 400) volume was 7 mL. The cellulose acetate membrane (0.45 µm, Sartorious) was kept in receptor media for 5 min right before applying the formulation. The applied formulation was 500 µL for each Franz’s diffusion cell. The sampling and replacing the receptor media volume were 500 µL for each time point. The test was performed on a vertical diffusion system (Teledyne Hanson Research, Chatsworth, CA, USA) with 6 cells under 32 °C system temperature. The sampling was performed for 8 h with sampling points at 0.5, 1, 2, 3, 4, 5, 6, and 8 h. The samples were analyzed on HPLC, and the percentage of cumulative released amount was calculated as a function of time.

#### 2.4.9. Cytotoxicity Studies

The cytotoxicity test was performed with a 3-[4,5-dimethylthiazol-2-yl]-2,5 diphenyl tetrazolium bromide (MTT) cytotoxicity assay based on the formation of formazan crystals via living cells. HaCaT keratinocyte (CLS 300493) was obtained from Cell Line Services (Heidelberg, Germany). The cell line was cultured at 37 °C and 5% CO_2_ atmosphere in Dulbecco’s Modified Eagles Medium supplemented with 10% fetal bovine serum, 1% streptomycin, and penicillin. After reaching the optimum cell density, the cells were seeded in a 96-well plate at 7 × 10^3^ cells/well density and treated with optimized CER-NP-loaded liposomes (100%, 50%, 25%, 12.5%, 6.25, and 3.12%), liposome without CER-NP as the negative control, and sodium lauryl sulfate (SLS) (5.0%, 2.5%, 1.25%) as the positive control. The exposure was maintained for 24 h.

After 24 h, the MTT solution (5 mg/mL in PBS) was applied to each plate after washing with 1× Dulbecco’s PBS. The cells were incubated for 2 h at 37 °C in an incubator in a humidified atmosphere of 5% CO_2_. The MTT and culture medium were removed, and formazan crystals were dissolved via dimethyl sulfoxide for 20 min. After dissolving, the absorbance of each well was recorded at 570 nm using a microplate reader (Thermo, Waltham, MA, USA). The cytotoxicity of the CER-NP-loaded liposomes and the controls were evaluated as a relative light unit (RLU) based on absorbance.

#### 2.4.10. Physical Stability Studies

The optimized CER-NP-loaded liposomes were kept at 25 ± 2 °C temperature and 60% relative humidity ± 5%, and the accelerated conditions were 40 ± 2 °C temperature and 75 ± 5% relative humidity during the 30 and 90 days with different samples [31]. The stability samples were analyzed in terms of their particle size, PDI, and zeta potential values to show their physical stability.

#### 2.4.11. Statistical Analysis

The results of each experiment were run in at least triplicate, and the data were then presented as the mean standard deviation (SD). One-way ANOVA was used for the statistical analysis along with Tukey’s post hoc test. A difference was deemed statistically significant if the *p*-value was less than 0.05.

## 3. Results and Discussion

### 3.1. Solubility of CER-NP

The in vitro release tests are performed based on the theory of drug diffusion, which takes the solubility of active molecules in receptor media as a prior condition. To maintain the sink condition, it is necessary to quantitatively determine the solubility of Cer-NP in alternative receptor media. The CER-NP is widely known in the literature for its aqueous solubility problem [32]. Therefore, the solubility test was performed to select the receptor media used in the in vitro release test for the CER-NP-loaded liposomes.

According to the CER-NP solubility test results, it showed low solubility in PBS pH 7.4. However, the scale-up and post-approval changes (SUPAC) guidelines allow the additive contents for poorly water-soluble drugs and ingredients [33]. The receptor medium alternatives and solubility of CER-NP are provided in Table 2.

The ethyl alcohol in R6 can just disperse the CER-NP grains based on the organoleptic evaluation. The BSA: PBS media is the most preferred one in drug formulation studies because of the hydrophilic and hydrophobic surfaces of BSA [34]. However, CER-NP was not solved in R5 media effectively. The R2 media have shown a peak on the chromatogram, but it was under the quantification limit in the HPLC method. The R3 and R4 give high solubility to CER-NP. Therefore, the R3 media were preferred as receptor media for the in vitro release test to maintain sink condition.

### 3.2. Experimental Design: CER-NP-Loaded Liposome Optimization

The optimization of CER-NP-loaded liposomes was carried out based on RSM in this study. The intercellular matrix in the *stratum corneum* consists of different ceramides, including CER-NP, fatty acids, and cholesterol, in a specific range. Through the RSM, CER-NP and other lipid substituents are used in harmony. The responses for the design were particle size and PDI value of liposomes. During the optimization process, the concentration of CER-NP was 2.19% of the total lipid amount, which is the value of lipid replacement therapy [35]. The oleic acid and cholesterol were in more than CER-NP molar concentrations to improve the release and diffusive properties of CER-NP. At this point, the RSM could have assisted the formulation optimization by specifying the lipid substituent amounts and maintaining the quality of CER-NP-loaded liposome.

Non-ionic surfactants such as Tween 80 and sodium cholate are widely used to form liposomes. However, destroying the orthorhombic lateral packing of the lipid barrier is observed due to applying a cosmetic formulation that includes ionic and non-ionic surfactants [36]. However, phosphatidylcholine needs surfactants to form a vesicle structure, even if this could be cholesterol. In this study, it is a priority to use cholesterol, which occurs in the skin’s extracellular matrix, as a surface-active agent in encapsulating the essential amount of CER-NP.

In the pre-formulation process, CER-NP and cholesterol-containing liposome formulations do not have a fine particle size, and the PDI value indicates the issue of the low solubility of CER-NP and membrane fluidity derived from cholesterol [37]. In the same manner, the presence of oleic acid changes the particle size and PDI value response in the pre-formulation study. However, the CER-NP-loaded liposomes should contain CER-NP, fatty acid, and cholesterol at an optimum ratio. With this design, the particle size response was minimized to accumulate the CER-NP-loaded liposome on the *stratum corneum*. The stability of CER-NP-loaded liposomes is typically associated with PDI values and zeta potential. The goal of this study was to minimize and reach a PDI value of less than 0.3 and a zeta potential of −15 < z < +15 mV [38].

### 3.3. Determination of Responses: Particle Size and PDI Value

In the optimization process, there are various models, such as quadratic, cubic, and factorial, to fit the response data in an efficient model. In our case, it was the most efficient model, quadratic for both particle size and PDI value based on the Design of Expert. The equation defining the quadratic model is provided in Equation (2).
Y: β_0_ + β_1_A + β_2_B + β_12_AB + β_11_A^2^ + β_22_B^2^(2)

Equation (2): Quadratic model equation for particle size and PDI value responses.

The predicted value is defined as Y in the equation. A and B are independent variables. β_1_ and β_2_ define linear coefficients; β_12_ represents interaction coefficients; β_11_ and β_22_ define quadratic coefficients. The formulation inputs about material attributes are listed in Table 3. During the optimization process, the formulations listed in Table 3 were prepared and examined to take the responses.

The second-order quadratic equation was used when minimizing the particle size. The predicted particle size was calculated via Equation (3), which is provided below.
PS: 180.61 − 16.01 A + 73.23 B − 70.46 C + 23.06 AB − 22.51 AC − 95.55B C + 3.64 A^2^ + 30.94 B^2^ + 57.53 C^2^(3)

Equation (3): Particle size responses equation based on quadratic model.

The statistical analysis of variances is provided in Table 4 for particle size. The particle size responses showed a good F-value (26.91) and low *p*-value (<0.0001), which means the equation of the model was significant. The lack-of-fit value was insignificant (*p* = 0.9384), which means that the model fitting is fine. The adjusted and predicted R^2^ values are 0.9247 and 0.8966, respectively. The values show that the experimental and predicted values are correlated.

The second-order quadratic equation provided below was used when minimizing the PDI value.
PDI: 0.57 − 0.18 A + 0.15 B − 0.28 C − 0.10 AB + 0.128 AC − 0.05 BC + 0.042 A^2^ + 0.039 B^2^ + 0.037 C^2^(4)

Equation (4): PDI value responses equation based on quadratic model.

The statistical analysis of PDI value responses is provided in Table 5. The PDI value responses show a good F-value (16.88) and low *p*-value (0.0001), indicating a significant model equation. The lack-of-fit value was not significant *(p* = 0.6827), which indicates fitting the model is fine. The adjusted and predicted R^2^ values are 0.8827 and 0.7587, respectively. The values show that the experimental and predicted values are correlated.

The desirability factor, which is crucial in optimization methods including multi-response parameters, was found to be 0.9387 in this optimization method. The value proves that the experimental value would meet the target of the predicted value.

According to the statistical analysis, CER-NP (factor A) is not significantly effective on both the particle size and PDI value (*p* = 0.7028 and 0.6571). However, cholesterol (factor B) and oleic acid (factor C) were significantly effective on the particle size and PDI value (*p* < 0.05). The one-factor analysis shows the individual effect of each factor on particle size and PDI values. The one-factor analysis is provided in Figure 2. Figure 2a,d shows the insignificant effect of CER-NP on particle size and PDI values. Figure 2b,e explains the point that the cholesterol increase raised the particle size and PDI value. The increase in oleic acid reduced the particle size and PDI value, as shown in Figure 2c,f.

Three-dimensional response surface graphs show the effects of the relations of ingredients. Figure 3a shows the effect of relations between oleic acid and cholesterol; meanwhile, CER-NP amount is maximized. Figure 3c indicates that oleic acid has a crucial effect on particle size in maximized cholesterol with any levels of CER-NP. The maximum amount of oleic acid can keep the particle size under control in many levels of cholesterol and CER-NP based on Figure 3e. Figure 3b shows that the PDI value was controlled by cholesterol in maximized CER-NP with any levels of oleic acid. However, the CER-NP and oleic acid may have an optimum ratio to obtain a minimum PDI value in the maximum level of cholesterol, according to Figure 3d. When oleic acid is maximized, an increasing amount of cholesterol can reduce the PDI value at a maximum level of CER-NP in Figure 3f.

The idea about raising particle size by high cholesterol amounts in the low level of oleic acid is supported by studies on liposomes containing cholesterol [39]. It is reported that an overrated amount of sterol causes an increase in the mean diameter of the liposome [40]. The increase in particle size might be related to interactions between lipid chains close to the head group of phosphatidylcholine and membrane stretching in the presence of sterols. Therefore, by increasing the cholesterol ratio, more than adequate cholesterol molecules may diffuse the phosphatidylcholine bilayer membrane. For this reason, the membrane fluidity is increased, and the close packing on the phosphatidylcholine bilayer membrane is prevented.

In addition, the study reports that the ceramide/stearic acid/cholesterol-based liposomal membrane shows that the disarrangement of molecules and phase softening occurred because of the increase in cholesterol concentration [41]. In our study, the oleic acid provides advantages on this point via increasing surface activity on liposomes. In this way, close packing on the phosphatidylcholine bilayer membrane could have been promoted. In the preformulation study, the impact of oleic acid on particle size, PDI value, and zeta potential was realized. The effect of cholesterol on particle size may be solved in the presence of oleic acid in liposomes. Kurniawan J. et al. showed that the oleic acid causes electrostatic repulsion on the dipalmitoylphosphatidylcholine membrane at neutral pH apart from low acidic pH [42]. Therefore, we thought that oleic acid could have improved the particle size by causing electrostatic repulsion and increasing the surface activity on the phosphatidylcholine bilayer membrane [42,43].

On the other hand, the high amount of oleic acid in a low level of cholesterol with maximum CER-NP can cause a rise in the PDI value. The rigidity derived from cholesterol provides stability to liposome structure, which is why cholesterol has a crucial role in PDI value. However, the cholesterol is not enough to obtain a fine PDI value without oleic acid based on preformulation studies (with oleic acid PS: 136.1, PDI: 0.248; without oleic acid, PS: 494.7, PDI: 0.706). Therefore, we thought that the PDI value brings up to the mark as the oleic acid/cholesterol ratio optimized.

#### Evaluation of Zeta Potential

Zeta potential responses were in the range between (−)1.85 and (−)10.3 mV that showed no significant difference statistically. Therefore, zeta potential has been descoped on responses in this study. Even so, oleic acid-containing liposomes have been negatively charged rather than the absence of oleic acid according to pre-formulation studies (with oleic acid ZP: −10.7 ± 1.11 mV; without oleic acid ZP: −1.30 ± 0.47 mV) (unpublished data).

### 3.4. Selection of the Optimized CER-NP-Loaded Liposomes

The optimized liposome formulation was selected based on minimizing the particle size and PDI value by the Design of Expert. The software suggested that the concentrations of CER-NP, oleic acid, and cholesterol are 2.4, 3.76, and 5%, respectively. In this hypothetical case, the particle size and PDI value would be 132.6 nm and 0.278, respectively.

The experimental particle size and PDI data of optimized liposomes, which performed at least three replications, were 136.6 ± 4.05 d.nm and 0.248 ± 0.012, respectively. As a result, the experimental responses were in close agreement with the predicted responses. The DLS result of the optimum CER-NP-loaded liposome is provided in Figure 4.

#### 3.4.1. Encapsulation Efficiency and CER-NP Content of the Optimized CER-NP-Loaded Liposomes

The optimum liposomes were evaluated due to their encapsulation efficiency. Considering the physicochemical structural similarity between ceramide subtypes and phosphatidylcholine, it might be possible to diffuse to CER-NP into the phosphatidylcholine bilayer membrane. Therefore, the selected CER-NP-loaded liposome results were provided in context. As a result, the CER-NP content was determined to be 97.54%. The encapsulation efficiency was 93.84 ± 0.87%. The high encapsulation efficiency could result in the increased lipophilicity of CER-NP.

#### 3.4.2. Confocal Laser Scanning Microscope Imagining of Optimized CER-NP-Loaded Liposomes

The optimized liposomes prepared with various molar concentrations of NBD-CER instead of CER-NP have been observed in CLSM. The fluorescent nitro group of NBD-CER is inevitably shown in a vesicle structure. The CLSM images are provided in Figure 5. The NBD-Cer amount was evaluated based on CLSM figures. Figure 5A contained the highest molar concentration of NBD-Cer. Figure 5B–D may contain less localized fluorescent areas rather than Figure 5A [44].

The localization of ceramides in liposomes is crucial in terms of the release characteristics of liposomes. This study focused on two preparation processes of ceramide-loaded liposomes, showing that ceramide is mainly encapsulated in a liposome bilayer unless there is another polymeric phase [45]. The CLSM imagining shows that the NBD-CER was successfully encapsulated in vesicles. In addition, we presume that the CER-NP was localized in the phosphatidylcholine layer apart from the aqueous inside of vesicles due to its lipophilicity. However, it is crucial to specify the NBD-Cer molar concentrations used instead of CER-NP in liposomes to obtain fine CLSM imagining. In the literature, the NBD-CER containing liposome was imagined under CLSM at various concentrations. Based on the authors’ view, the NBD-CER was localized on the bilamellar membrane of the liposome [30]. This consideration supports the idea that CER-NP and other lipid substituents might be composed in harmony within the phosphatidylcholine bilamellar membrane.

#### 3.4.3. ATR-IR Analysis of Optimized CER-NP-Loaded Liposomes

The NBD-CER has a specific 1,2,5 oxadiazole group on its chemical structure. In addition, the oxadiazole ring is bonded to 4-nitroaniline and becomes a nitrobenzoxadiazole (-NBD) functional group as fluorescent functionality. The chemical structure and functional group of NBD-CER are provided in Figure 6.

The liquid phase of optimized liposomes containing NBD-CER was analyzed via ATR-IR spectroscopy in comparison with the liquid dispersion of empty liposomes based on transmittance (%). The spectral data are provided in Figure 7. According to the results, the C-N stretch was shown up in 1224.80 cm^−1^ as amide III in liquid disperse liposome with NBD-CER (red). The specific regions were seen in only liquid disperse liposomes with NBD-CER (red), not in liquid disperse empty liposomes (blue) as compared to the spectral data.

In addition, there are other characteristic peaks of liposomes in ATR-IR. The PO_2_ symmetric stretching region coming from phosphatidylcholine is shown at 1082.07 cm^−1^ in lyophilized liposomes with CER-NP (black) [46,47,48]. This region gives an idea about the microenvironment of PO_2_^−^ groups in liposomes. The sharp peak on 1737.86 cm^−1^ on the phosphatidylcholine spectrum is derived from the non-hydrogen bonded C=O group. However, in aqueous media, water may diffuse into the side of the carbonyl group on the lipid bilayer and make hydrogen bond interaction on the carbonyl group area [49,50,51,52]. Therefore, the sharp peak may split in aqueous dispersion because of the lipid–water interface on the side of the carbonyl group [47,53]. The CH_2_ symmetric stretch and CH_2_ asymmetric stretch were shown at 2854.65 cm^−1^ and 2924.09 cm^−1^ in liquid-dispersed liposomes with NBD-Cer (red). In many studies, the CH_2_ vibration at 2852–2855 cm^−1^ is considered a sign of a liquid crystalline phase that leads to liposome formation [54,55,56]. These are derived from the characteristics of lipid acyl chain packing and fluidity on liposomes [53,57]. In lyophilized liposomes with CER-NP (black), the CH_2_ vibration region shows a lower frequency at 2852.72 cm^−1^. This indicates a more regular and ordered lipid organization, which is like orthorhombic lateral packing of skin lipids in a healthy stratum corneum. Le Deygen I.M. et al. reported the specific peaks of liposomes in ATR-IR analysis [47]. Stretching vibrations of the CH_2_ group are symmetric and asymmetric, stretching into 2850 cm^−1^ and 2919 cm^−1^ bands, which are like our findings.

#### 3.4.4. FE-SEM Imagining of Optimized CER-NP-Loaded Liposomes

The lyophilized CER-NP-loaded liposomes dispersed in mannitol as cryoprotectant are shown as specific liposome images [58]. The spherical feature of liposomes is observed in FE-SEM images (Figure 8). The lyophilized CER-NP-loaded liposomes smaller than 500 nm were indicated with blue arrows on the images. Figure 8A,B shows the close view of liposomes (Mag: 25,000×). Figure 8C,D shows the distribution of liposomes in a wide perspective on a small scale (Mag: 10,000×). In electron microscopy, it is well known that high voltage can degrade the samples. The FE-SEM principle minimizes the voltage applied on the sample surface. In this way, the liposomes are observed under low kilovolts without any degradation and no need for sample covering. The surface morphology of liposomes was observed in 3D and real time in FE-SEM. The morphology of liposomes was hexagonal, similar to those observed in the literature [59]. The small-size liposomes are mostly dispersed in mannitol. The particle sizes and FE-SEM images are also comparable to those reported in the literature [59,60].

#### 3.4.5. Physical Stability Studies

The physical stability results of optimized CER-NP-loaded liposomes are provided in Table 6. The stability was evaluated for 3 months based on particle size and PDI value, which are crucial characteristics indicating physical liposome stability. During the 30 days, the optimum CER-NP-loaded liposomes were observed in monophasic features at 25 °C temperature and 60% relative humidity. However, the liposomes could not keep the monophasic feature under accelerated stability conditions even for 30 days. The DLS results are provided in Table 6.

Based on the DLS result, the formulation shows a good stability property for the first 30 days period at 25 °C temperature and 60% relative humidity. The main point might be the zeta potential provided by oleic acid. However, the fine zeta potential is not all issues. The cholesterol in optimized ratio might have an impact on PDI values during the 30 days, but it is well-known that dispersed systems are not suitable for long-term stability without viscosity-increasing excipients. The phospholipids’ transition temperature (Tc) has a pivotal role in the stability of the liposomal system [61]. In our study, Lipoid S100 has an approximate Tc value with the temperature condition of the long-term stability test [62]. The particle size and PDI values increased comprehensibly under the 25 °C temperature conditions. However, the particle size and PDI value increased dramatically under the 40 °C temperature conditions. This state might explain why the liposomal system is dramatically unstable under accelerated stability test conditions.

#### 3.4.6. In Vitro CER-NP Release from Optimized Liposomes

It is well-known that liposomes provide a sustained drug release via lipid bilayer. Therefore, the in vitro CER-NP release from liposomes is crucial to observe the CER-NP release rate as a function of time. In vitro release of CER-NP from optimized liposomes was evaluated based on the cumulative released percentage of CER-NP from liposomes. As was seen in Figure 9, more than 50% of CER-NP was released from the optimum liposome until the 6th hour. Almost the entire amount of the CER-NP (89%) was released at the end of the 8 h. The release rate was calculated at 0.54 ± 0.013 (k, µg.cm^−2^·h^−1^).

In general, non-linear regression models are commonly preferred to express the release profile of liposomes. According to the results, the cumulative release may fit the Korsmeyer–Peppas kinetic model. The Korsmeyer–Peppas equation (M_t_/M_∞_ = kt^n^) shows an exponent value between 0.5 < n < 1.0, which means the formulation expresses a non-Fickian diffusion profile [63,64]. The n value is close to 1.0, and the drug release rate is independent of time. In Figure 10, the regression coefficient of the Korsmeyer–Peppas model was over 1.0, which means the release profile is interpreted as close to the super case II transport drug release mechanism.

In the literature, there are many cases in which the drug release from liposomes fitted the Korsmeyer–Peppas kinetic model [63,65,66]. The CER-NP release from optimized liposomes showed a similar release pattern as reported in the literature.

#### 3.4.7. Cytotoxicity Analysis of Optimized CER-NP-Loaded Liposomes

Even the CER-NP, oleic acid, and cholesterol are naturally bio-transformed through the skin surface; the concentration of skin lipids in the formulation can cause skin irritation. To work the safety assessment, the liposomes without CER-NP and the optimized CER-NP-loaded liposomes were applied to the keratinocyte cell line in comparison with SLS as positive control, which is acceptable as cytotoxic on human keratinocytes within each concentration used in this study [67,68].

As a result, the relative light unit on 100% concentration of the optimized CER-NP-loaded liposomes was 6.89 times higher than those of the highest concentration of SLS. The results are provided in Figure 11. This signifies that the optimized CER-NP-loaded liposomes could be considered relatively safe for lipid replacement therapy on AD via topical administration.

In addition, the optimum CER-NP-loaded liposomes had just 0.63 times lower RLU than those without CER-NP. This means there may be no concern about the CER-NP content used in this study.

## 4. Conclusions

In recent years, the lipid replacement approach via liposomes has reached the trend of restoring the skin barrier. The skin lipids feature low formulation-forming capacity due to being poorly soluble in excipients used in topical dosage forms. In this study, CER-NP-loaded liposomes were designed and optimized by using RSM. According to the experimental design, the optimum CER-NP-loaded liposomes were obtained at a fine particle size and PDI value. The central composite design presents a prediction model that showed a good correlation with experimental responses. The CLSM images support the encapsulation of CER-NP in liposomes. The optimized liposomes were obtained with high CER-NP content and encapsulation efficiency. ATR-IR analysis supported the characteristic liquid crystalline phase formation of the liposome. The liposomes were physically stable in terms of particle size and distribution. The release profile of the liposomes was fitted with the Korsmeyer–Peppas model. The cytotoxicity studies showed that the CER-NP-loaded liposomes could be considered relatively safe.

In conclusion, the positive and negative sides of lipid substituents were harmonized via RSM. It is clearly understood that each lipid substituent in CER-NP-loaded liposomes is a cornerstone for the quality of skin products. The liposome optimization process is performed cost-effectively, and the optimized CER-NP-loaded liposomes to potentially restore the skin barrier function was obtained. Nevertheless, to confirm the efficacy of the optimized liposome formulation, further skin penetration and clinical efficacy data should be obtained.

## Figures and Tables

**Figure 1 pharmaceutics-15-02685-f001:**
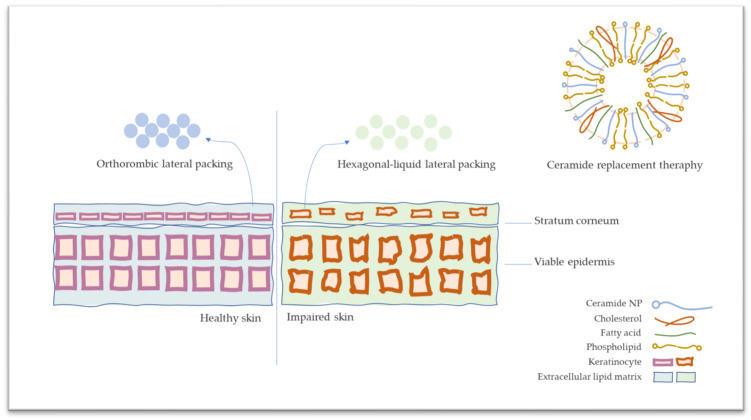
The schematic presentation of stratum corneum on healthy and atopic dermatitis skin and ceramide replacement therapy.

**Figure 2 pharmaceutics-15-02685-f002:**
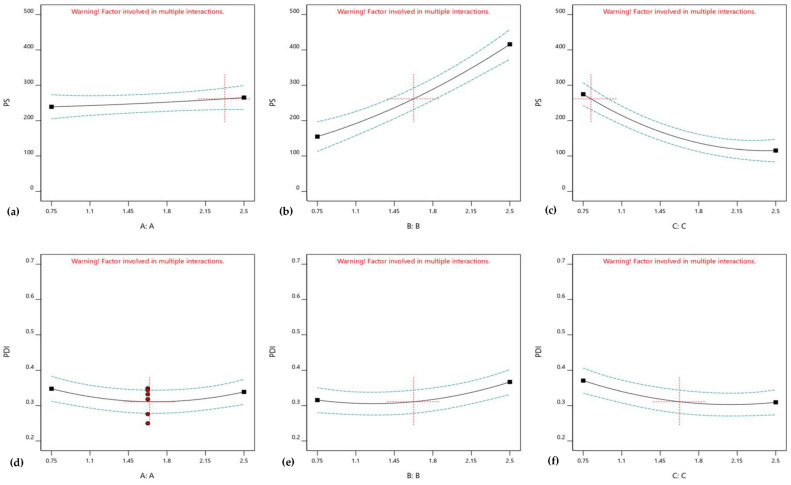
One-factor analysis of CER-NP (**a**,**d**), cholesterol (**b**,**e**), and oleic acid (**c**,**f**) on CER-NP-loaded liposome responses; particle size (**a**–**c**) and PDI value (**d**,**f**,**g**).

**Figure 3 pharmaceutics-15-02685-f003:**
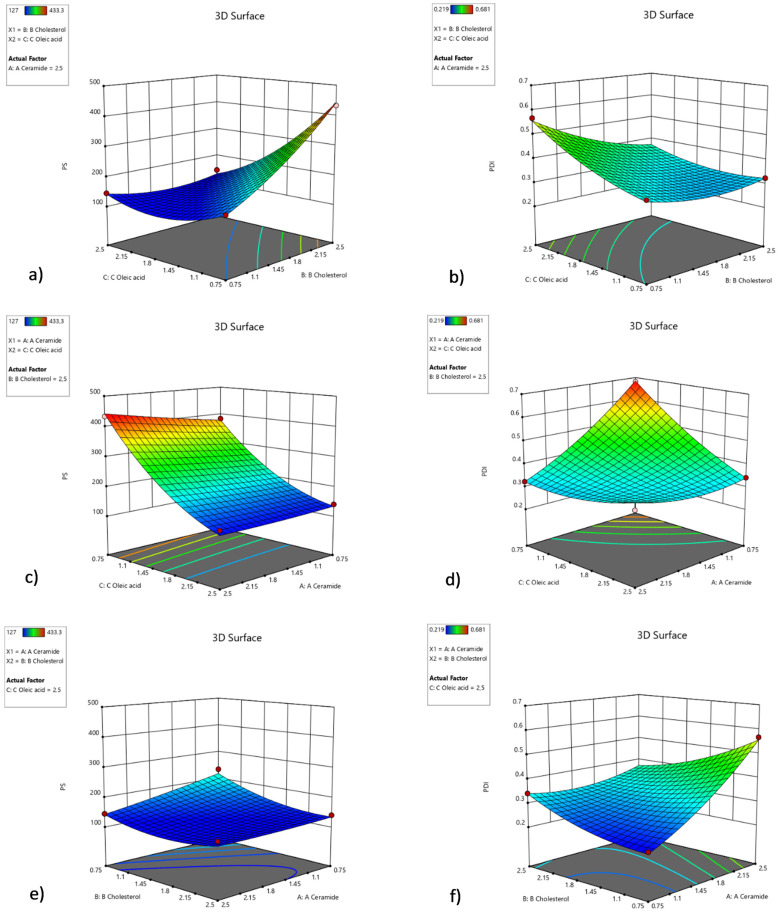
Three-dimensional response surface graph involving factors’ interactions with each other in optimization. The interaction of cholesterol and oleic acid factors within the high amount of CER-NP on particle size (**a**) and PDI value (**b**), the interaction of oleic acid and CER-NP factors within the high amount of cholesterol on particle size (**c**) and PDI value (**d**), the interaction of cholesterol and CER-NP factors within the high amount of oleic acid on particle size (**e**) and PDI value (**f**).

**Figure 4 pharmaceutics-15-02685-f004:**
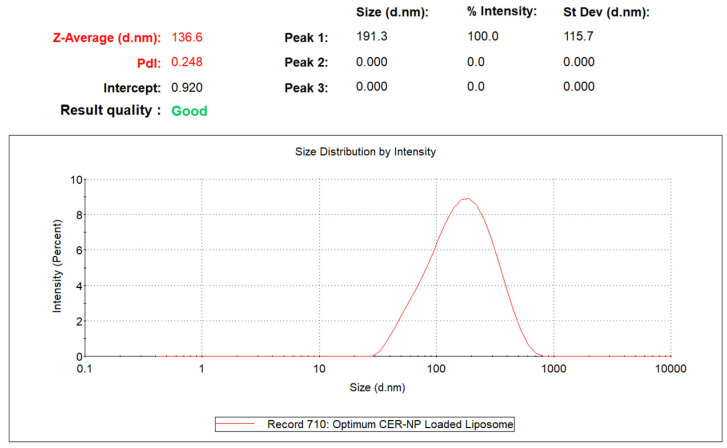
The particle size and PDI value of optimized CER-NP-loaded liposomes.

**Figure 5 pharmaceutics-15-02685-f005:**
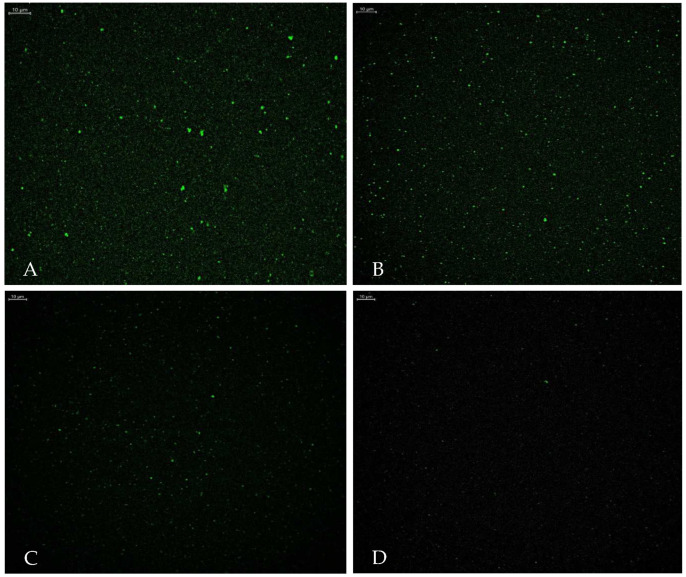
Confocal laser scanning microscope images of optimum liposome with NBD-Cer at (**A**) 1.73‰ mmol, (**B**) 1.39‰ mmol, (**C**) 0.68‰ mmol, and (**D**) 0.34‰ mmol NBD-Cer (Scale bar: 10 µm).

**Figure 6 pharmaceutics-15-02685-f006:**
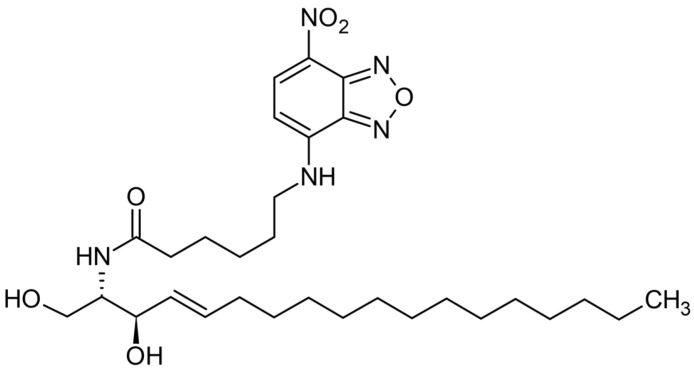
Chemical structure and functional group of NBD-Cer.

**Figure 7 pharmaceutics-15-02685-f007:**
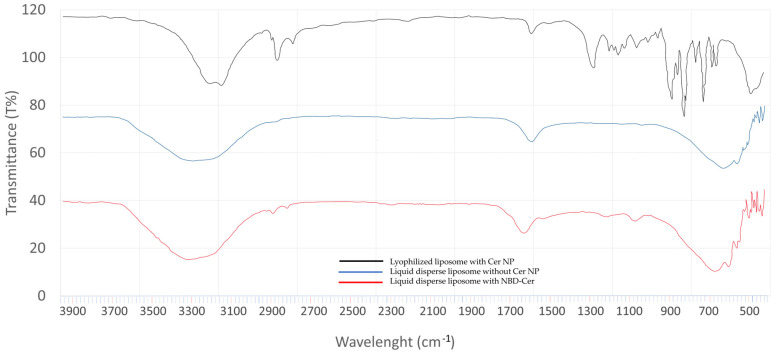
Spectral data of lyophilized liposome with CER-NP (black), liquid dispersion of liposome with CER-NP (blue), and liquid dispersion of liposome with NBD-CER (red).

**Figure 8 pharmaceutics-15-02685-f008:**
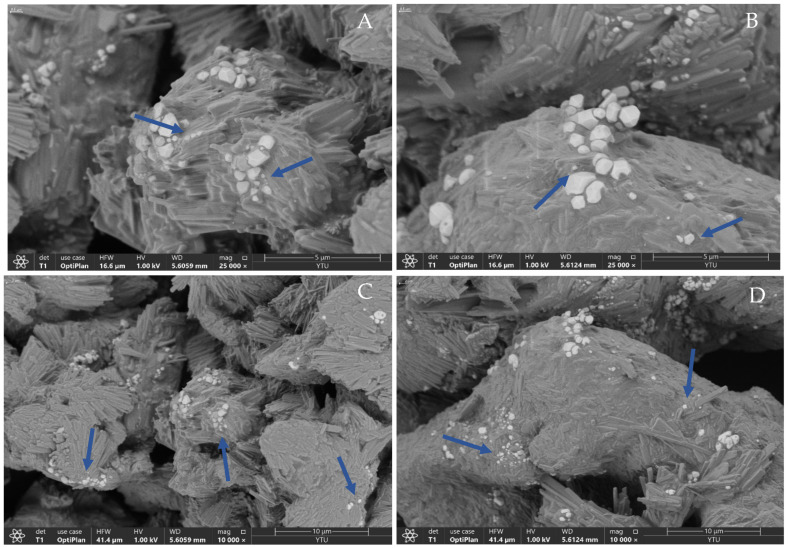
The FESEM image of optimum CER-NP-loaded liposomes in large scale ((**A**,**B**); mag: 25,000) and small scale ((**C**,**D**); mag: 10,000). The blue arrows indicated the lyophilized CER-NP-loaded liposomes.

**Figure 9 pharmaceutics-15-02685-f009:**
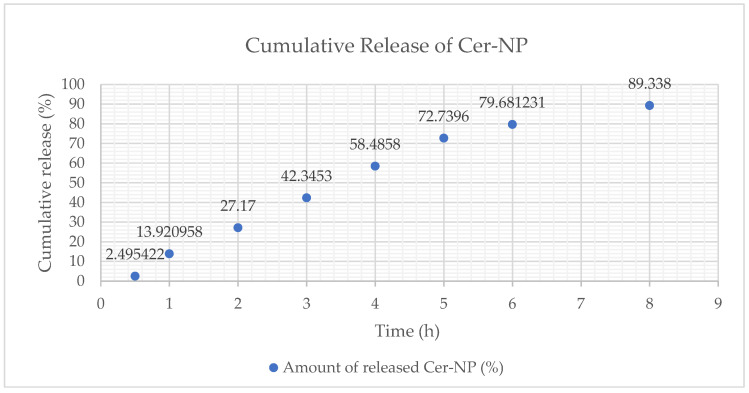
The cumulative release (%) profile of CER-NP from optimum liposomal formulation, cumulative release percentage (y), and time point (x).

**Figure 10 pharmaceutics-15-02685-f010:**
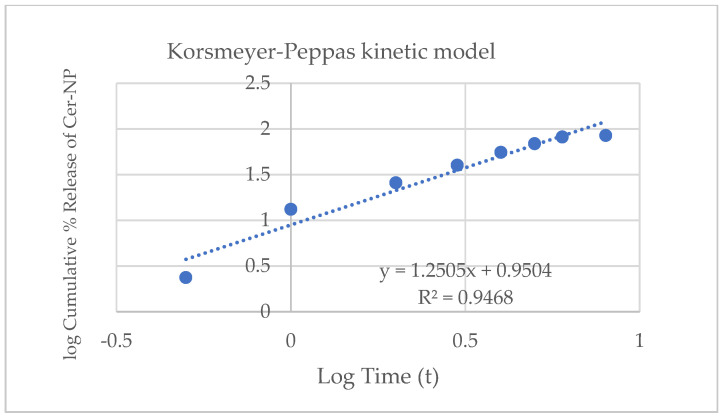
Release kinetic of optimum liposomal formulation fitting on Korsmeyer–Peppas kinetic model.

**Figure 11 pharmaceutics-15-02685-f011:**
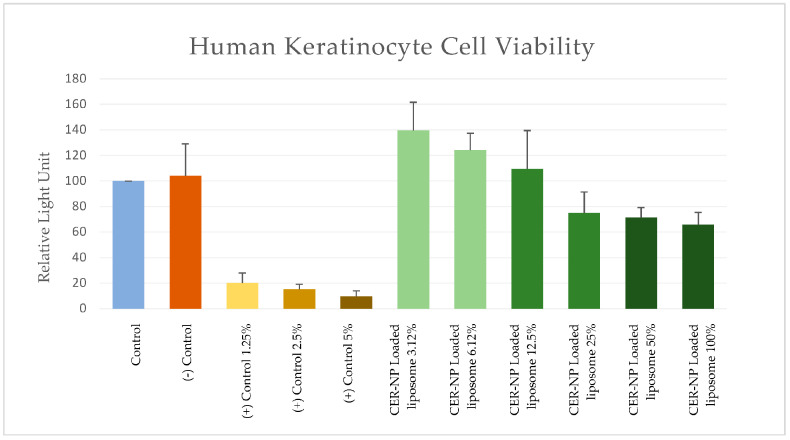
MTT assay; RLU values of each group as cell viability sign on cytotoxicity tests. Control is a non-treatment, (−) control is a liposome without CER-NP, and (+) control is SLS at distinct concentrations. Optimum CER-NP-loaded liposome series are obtained via dilution. The final CER-NP molar concentration is provided below the figure.

**Table 1 pharmaceutics-15-02685-t001:** Selected variables in central composite design.

			Levels of Variable			
	Factors	−α	−1	0	+1	+α
A	CER-NP (mg)	0.15	0.75	1.6	2.5	3.09
B	Cholesterol (mg)	0.15	0.75	1.6	2.5	3.09
C	Oleic acid (mg)	0.15	0.75	1.6	2.5	3.09

The amounts of each lipid are referred to as mg/50 mg phospholipid content.

**Table 2 pharmaceutics-15-02685-t002:** CER-NP solubility in various in vitro release test receptor media.

Receptor Medium	Ingredients	Solubility (µg/mL)	Evaluation
R1	PBS	Not soluble	−
R2	PBS 10% propylene glycol	Not detectable	−
R3	PBS 10% PEG 400	224.45	+
R4	PBS 2% Tween80	71.5	−
R5	PBS 1% BSA	Not detectable	−
R6	PBS 25% ethyl alcohol	Not detectable	−

ND: Not detected, under the quantification limit.

**Table 3 pharmaceutics-15-02685-t003:** The variable lipid amounts (mg) per phospholipid amounts (50 mg) and particle size and PDI value results of formulations.

Formulation	CER-NP	Cholesterol	Oleic Acid	Particle Size	PDI
L1	1.624	1.57	0.13	396	0.428
L2	2.43	0.77	2.45	144.8	0.567
L3	2.35	0.73	0.78	164	0.349
L4	3.06	1.62	1.62	152.7	0.405
L5	1.66	3.144	1.687	177.4	0.406
L6	1.62	1.61	1.6	177.4	0.349
L7	1.62	1.63	1.6	210.3	0.45
L8	0.72	0.75	2.47	234.5	0.519
L9	0.75	2.4	0.77	463.3	0.681
L10	2.47	2.6	2.47	144.1	0.343
L11	1.61	1.61	1.62	130.4	0.276
L12	0.16	1.62	1.63	155.4	0.389
L13	1.61	1.61	1.61	127	0.344
L14	1.61	0.16	1.63	131.9	0.318
L15	0.71	0.72	0.76	162.1	0.429
L16	2.41	2.45	0.71	433.3	0.323
L17	0.75	2.42	2.4	140.5	0.341
L18	1.61	1.64	1.6	133.9	0.332
L19	1.64	1.62	1.65	129.2	0.301
L20	1.6	1.62	3.05	145.5	0.348

**Table 4 pharmaceutics-15-02685-t004:** Statistical analysis of particle size response in ANOVA.

Source	Sum of Squares	df	Mean Square	F-Value	*p*-Value	
Model	1.735 × 10^−5^	9	19,276.85	26.91	<0.0001	significant
A-A	110.47	1	110.47	0.1542	0.7028	
B-B	32,790.64	1	32,790.64	45.77	<0.0001	
C-C	59,335.80	1	59,335.80	82.83	<0.0001	
AB	2495.71	1	2495.71	3.48	0.0915	
AC	2377.05	1	2377.05	3.32	0.0985	
BC	42,822.01	1	42,822.01	59.77	<0.0001	
A²	112.48	1	112.48	0.1570	0.7002	
B²	8087.01	1	8087.01	11.29	0.0072	
C²	27,968.09	1	27,968.09	39.04	<0.0001	
Residual	7163.91	10	716.39			
Lack of Fit	1296.88	5	259.38	0.2210	0.9384	not significant
Pure Error	5867.03	5	1173.41			
Cor Total	1.807 × 10^−5^	19				

Note: R^2^ = 0.9603 Adj. R^2^ = 0.9247 Pred. R^2^ = 0.8966.

**Table 5 pharmaceutics-15-02685-t005:** Statistical analysis of PDI value response in ANOVA.

Source	Sum of Squares	df	Mean Square	F-Value	*p*-Value	
Model	0.1984	9	0.0220	16.88	<0.0001	significant
A-A	0.0003	1	0.0003	0.2093	0.6571	
B-B	0.0101	1	0.0101	7.73	0.0194	
C-C	0.0146	1	0.0146	11.18	0.0074	
AB	0.0487	1	0.0487	37.28	0.0001	
AC	0.0776	1	0.0776	59.45	<0.0001	
BC	0.0134	1	0.0134	10.30	0.0093	
A²	0.0150	1	0.0150	11.46	0.0069	
B²	0.0132	1	0.0132	10.12	0.0098	
C²	0.0122	1	0.0122	9.31	0.0122	
Residual	0.0131	10	0.0013			
Lack of Fit	0.0051	5	0.0010	0.6386	0.6827	not significant
Pure Error	0.0080	5	0.0016			
Cor Total	0.2114	19				

Note: R^2^ = 0.9382 Adj. R^2^ = 0.8827 Pred. R^2^ = 0.7587.

**Table 6 pharmaceutics-15-02685-t006:** The PS and PDI results, which are obtained at different time points at 25 °C and 40 °C, refer to the stability of the optimal CER-NP-loaded liposome.

Time (Day)	Particle Size (d.nm)		PDI	
	25 °C	40 °C	25 °C	40 °C
0	136.6 ± 4.05	-	0.248 ± 0.012	-
30	140.5 ± 3.38	181.0 ± 44.3	0.260 ± 0.012	0.369 ± 0.02
90	157.1 ± 1.53	215.4 ± 55.7	0.339 ± 0.05	0.613 ± 0.33

## Data Availability

Not applicable.

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
