# Peer review of "The Design and Optimization of Ceramide NP-Loaded Liposomes to Restore the Skin Barrier"

_pharmaceutics, 2023, doi:10.3390/pharmaceutics15122685_

Round 1
Reviewer 1 Report
Comments and Suggestions for Authors
The paper deals with an interesting subject, which is the design and optimization of ceramide NP loaded liposomes to restore skin barrier. The paper is of clear interest for the reader. It is well written and referenced. The quality of the presented data is high level. All the experimental design, from the optimization and modeling is well done, presented and described. Nevertheless, the paper is not suitable for publication in the present form and some major changes should be properly addressed by the authors to improve the clarity and the understanding.
- The authors should improve the English editing.
- In the title as well as in the text it’s not clear the use of “NP”.
- The authors should double check all the used acronyms and describe them the first time they appear.
- The authors should improve the quality of Figure 1, 4, and 5.
- In Figure 5, scale bar is missing.
- The authors should double check the numbers of Figures and if they are correctly cited in the text. For example, Figure 2, page 14, should be Figure 7.
- The authors should better describe 8, what the blue arrows indicate? Why did the authors report 4 images? The authors should enrich the figure caption, with more details.
- I would suggest the authors to organize the figures in panels, to better attract the attention and simplify the understanding of the reader.
- The authors should report a statistical analysis of the data.
Comments on the Quality of English LanguageThe authors should improve the English editing, some minor mistakes can be easily corrected.
Reviewer 2 Report
Comments and Suggestions for Authors
The submitted manuscript entitled Design and Optimization of Ceramide NP Loaded Liposomes to Restore Skin Barrier is dealing with the development of an optimized liposomal formulation based on response surface methodology (RSM).
General comments
The authors focus on optimized and balanced composition of liposomes as potential lipid replacement therapy against AD. In the text they describe the procedure to optimize the intended liposomes, which should correspond with the natural composition of CER-NP, Cholesterol and Oleic acid.
As Phospholipid they used Phospholipon S100 from Lipoid. On the Lipoid online ordering platform, which is available for me, Phospholipon S100 is not listed, please check this fact to identify which Phospholipid is exactly used.
Interestingly, the authors mainly focus on the components which should be entrapped and less attention is paid on the phospholipids, which are the membrane skeleton of the vesicles (figure 1). Why is the predicted position of the two other components not depicted
The phospholipid composition as well as the molar ratios may also affect the quality and function of the liposomes. The authors should explain why they did not consider the relevance of the core lipids, although it would be quite simple to extend the model by the phospholipids.
As Indicators for the RSM based optimization mean size, PDI and zeta potential were identified, however it should be explained why not also release and cytotoxicity?
The experimental design is in detail explained however the provided tables are partially incomplete and should be revised, see comments below.
Detailed comments
Materials and methods
2.1. Materials: is the correct name of the phospholipid Lipoid S100? In my Lipoid ordering list Phospholipon S100 is not provided
2.2. Preparation Method of CER-NP Loaded Liposomes : How are the empty liposomes composed – only Phospholids?
2.3. Experimental Design of CER-NP-loaded Liposomes : Table 1 is not completed- are the individual values factors, amounts…?
2.4.2. Determination of HPLC method of CER-NP in Liposomes: Which detector is used? The retention time was 9.27 min – for with substance. 3 different compounds are relevant, how are they quantified?
2.4.4. Confocal Laser Scanning Microscopy Imagining of CER-NP-loaded Liposomes: The fluorescent liposomes were prepared with various ratios of NBD-CER (1.73, 1.39, 0.68, 0.34, 0.17‰ mmol) what does that refer to?
Results and discussion
3.1. Solubility of CER-NP: IVRT in full text could not be found; For which purpose is the IVRT performed? It will not predict the release in vivo, aim of the experiment should be explained
3.2. Experimental Design: CER-NP-Loaded Liposome Optimization: Table 3 - Composition of the liposomes - mg/50mg - what does that refer to? What is meant by pre-formulation process? Values of size and PDI respectively standard deviation is missing.
3.4. Selection of The Optimized CER-NP-Loaded Liposomes: What is the rationale for the optimized liposome specification. In table 3 several formulations would have an appropriate size and the PDI is close to the specified value of 0.3. Repeated measurement could change the selection thus more than one formulation would be convenient.
Figure 4: Did the authors repeat these measurements, the graph shows an asymmetric peak and a quite broad distribution. May this heterogeneity interfere with the in vivo release and penetration.
3.4.2. Confocal Laser Scanning Microscope Imagining of Optimized CER-NP Loaded Liposomes: It can be assumed that CER-NP is incorporated or strongly associated with the core membrane. In this respect it remains still open were exactly the CER-NP is positioned. To prove the incorporation more than one fluorophore and better resolution is required. Figure 8 indicates surface associated CER-NP, the authors should discuss this circumstance in more detail. Legend of figure 8 is incomplete.
What exactly is shown in Table 6: 2 measurements at 25°C and 2 at 40°C – same or different samples ?
3.4.6. In vitro CER-NP Release from Optimized Liposomes: As mentioned above - How relevant is this result for the intended application.
Conclusion
The aim of the authors to develop a liposomal formulation with improved quality based on RSM is an excellent concept, due to cost reasons and experimental effort. However, for the submitted manuscript I recommend to thoroughly revise the procedures as well as their conclusions/decision arguments to substantiate the quality of this manuscript.
Comments on the Quality of English Language
Wording is in general ok, typing errors should be corrected
Round 2
Reviewer 2 Report
Comments and Suggestions for Authors
Dear Authors,
thank´s a lot for the positive response.
All remaining questions are well answered and the manuscript is revised accordingly. Additional information, which is now provided in the present document improve the overall quality of the manuscript.
Comments on the Quality of English LanguageEnglish seems to be fine, typing errors and italic writing should be checked